Manuscript prepared for Earth Surf. Dynam.
with version 2015/04/24 7.83 Copernicus papers of the LaTeX class copernicus.cls.
Date: 30 March 2016

# A nondimensional framework for exploring the relief structure of landscapes

Stuart W. D. Grieve[1], Simon M. Mudd[1], Martin D. Hurst[2], and David T. Milodowski[1]

[1]School of GeoSciences, University of Edinburgh, Drummond Street, Edinburgh EH8 9XP, UK
[2]British Geological Survey, Keyworth, Nottingham NG12 5GG, UK

*Correspondence to:* Stuart W. D. Grieve (s.grieve@ed.ac.uk)

**Abstract.** Considering the relationship between erosion rate and the relief structure of a landscape within a non-dimensional framework facilitates the comparison of landscapes undergoing forcing at a range of scales, and allows broad scale patterns of landscape evolution to be observed. We present software which automates the extraction and processing of relevant topographic parameters to rapidly generate non-dimensional erosion rate and relief data for any landscape where high resolution topographic data are available. Individual hillslopes are identified using a connected components technique which allows spatial averaging to be performed over geomorphologically meaningful spatial units, without the need for manual identification of hillslopes.

The software is evaluated on four landscapes across the continental United States, three of which have been studied previously using this technique. We show that it is possible to identify whether landscapes are in topographic steady state. In locations such as Cascade Ridge, CA a clear signal of an erosional gradient can be observed. In the Southern Appalachians, non-dimensional erosion rate and relief data are interpreted as evidence for a landscape decaying following uplift during the Miocene. An analysis of the sensitivity of this method to free parameters used in the data smoothing routines is presented which allows users to make an informed choice of parameters when interrogating new topographic data using this method. A method to constrain the critical gradient of the nonlinear sediment flux law is also presented which provides an independent constraint on this parameter for three of the four study landscapes.

## 1 Introduction

The Earth's surface evolves dynamically in response to the interplay of climatic, tectonic and other factors operating at timescales ranging from minutes to millenia. High resolution topographic data generated from terrestrial and airborne laser scanning, in combination with increased computational power has facilitated a revolution in geomorphology, allowing the quantitative interrogation of landscape form to provide insight into the forces shaping a landscape. Relationships have been found between topography and the tectonic (e.g., Wobus et al., 2006; Hilley and Arrowsmith, 2008; DiBiase

et al., 2012; Hurst et al., 2013a), climatic (e.g., Gabet et al., 2004; Anders et al., 2008; Champagnac et al., 2012) and biotic (e.g., Roering et al., 2010; Milodowski et al., 2015a) forcing of a landscape in addition to links between topography and bedrock properties (e.g., Korup, 2008; Clarke and Burbank, 2010, 2011; Hurst et al., 2013b).

Such fundamental relationships provide important insight into landscape evolution, however many of these techniques are challenging to implement, due to variable or poorly defined methods, or require proprietary software to obtain data. This highlights the need for standardized techniques and tools to allow the analysis of topographic data to be reproduced and falsified, strengthening our understanding of the processes that shape planetary surfaces. In this contribution we focus on

methods exploiting high resolution topographic data in soil mantled landscapes that aim to elucidate both sediment flux laws (c.f., Dietrich et al., 2003) and the transient evolution of landscapes (e.g., Hurst et al., 2013a).

     Our approach is rooted in a non-dimensional framework that describes relationships between erosion rates and hillslope topography in soil mantled landscapes (Roering et al., 2007). This frame-

work facilitates the direct comparison of landscapes of widely varying morphology and process. It has been shown to provide compelling insight into the identification of landscape transience (Hurst et al., 2012), complex tectonic signals from topography (Hurst et al., 2013a), and process controls on the density of channels (Sweeney et al., 2015). Extracting the nondimensional parameters from high resolution topography can be difficult, subject to choices about how the metrics are calculated,

and there has been no investigation into how different methods might influence results, and therefore the interpretation of landscapes.

     Here we present a framework and methodology for extracting the required topographic parameters and processing the resulting data. Our software uses a clear methodology to allow researchers to generate these data for new landscapes and can replicate published relationships between non-

dimensional erosion rate and relief. Such relationships can be used to discriminate between landscapes in topographic steady state, where erosion rate is balanced by uplift rate, and those undergoing transience or topographic decay.

     Additionally we present a method for generating spatially contiguous hilltop patches, required as a spatial averaging tool in many studies (e.g., Perron et al., 2009; Hurst et al., 2012, 2013a) to

identify individual hillslopes for analysis. An analysis on the influence of spatial averaging and data smoothing on the interpretation of topographic data is undertaken and hillslope and basin average data are also used to estimate the critical gradient, a key parameter in the nonlinear sediment flux model.

## 2  Theoretical Background

Numerous sediment flux laws (cf. Dietrich et al., 2003) have been developed and tested, particularly since the advent of cosmogenic radionuclide dating and high resolution topographic measurements. In addition to the conceptually simple linear flux law (Culling, 1960; McKean et al., 1993; Tucker and Slingerland, 1997; Small et al., 1999; Booth et al., 2013), models of depth dependent (Braun et al., 2001; Furbish and Fagherazzi, 2001; Heimsath et al., 2005; Roering, 2008) and nonlinear

sediment flux (Andrews and Bucknam, 1987; Roering et al., 1999, 2001, 2007) have been employed, alongside models which directly consider sediment particle motion (Foufoula-Georgiou et al., 2010; Tucker and Bradley, 2010; Furbish and Roering, 2013).

Models which consider particle motion are challenging to apply to real topography as they do not have an analytical solution and without high resolution soil depth information it is challenging

to apply a soil thickness based sediment flux law to landscape scale analysis (Grieve et al., 2016). However, topographic predictions of the nonlinear flux law have been successfully tested (Roering et al., 2007; Grieve et al., 2016) suggesting that it, at a minimum, can constrain broad scale sediment transport processes across landscapes. The nonlinear flux law is (Andrews and Bucknam, 1987; Roering et al., 1999, 2001, 2007),

$$\bar{q}_s = \frac{KS}{1 - (|S|/S_c)^2}, \tag{1}$$

where $S$ is the topographic gradient in dimensions of Length/Length (dimensions denoted in square brackets as [$L$]ength, [$M$]ass and [$T$]ime), $S_c$ [dimensionless] is the hillslope critical gradient, $K$ [$L^2T^{-1}$] is a sediment transport coefficient and $\bar{q}_s$ [$L^2T^{-1}$] is a volumetric sediment flux per unit

contour length. As $S$ tends towards $S_c$, the sediment flux asymptotically increases towards infinity, corresponding to an increase in landsliding on an increasingly planar hillslope.

Roering et al. (2007) modeled the relief structure of theoretical one-dimensional hillslopes which evolve under Equation 1 and found that relief, the difference in elevation between a hilltop and the point on the channel it is coupled to, is controlled by the erosion rate, hillslope length and

the sediment transport coefficient. Equation 1 has been found to be consistent with observations of topography and erosion rates across several landscapes (e.g., Roering et al., 1999, 2007; Roering, 2008; Hurst et al., 2012)

Roering et al. (2007) normalized relationships describing these one-dimensional hillslopes using topographic parameters to produce a dimensionless erosion rate,

$$E^* = \frac{E}{E_R} = \frac{\rho_r}{\rho_s} \cdot \frac{2EL_H}{KS_c} = \frac{-2C_{HT}L_H}{S_c} \tag{2}$$

where $E$ $[LT^{-1}]$ is the erosion rate, $\rho_r$ and $\rho_s$ $[ML^{-3}]$ are the rock and soil bulk densities, respectively, $C_{HT}$ $[L^{-1}]$ is the hilltop curvature, $L_H$ $[L]$ is the hillslope length, $E_R$ $[LT^{-1}]$ is a reference erosion rate denoted as:

$$E_R = \frac{KS_c}{2L_H(\rho_r/\rho_s)},$$ (3)

and the dimensionless relief is given as:

$$R^* = \frac{R}{S_c L_H},$$ (4)

where R $[L]$ is the topographic relief. Parabolic hillslope profiles are generated when $E^*$ values are less than or equal to one, such that $R^*$ increases approximately linearly with erosion rate. Planar hillslopes near the critical gradient, $S_c$, indicate that $R^*$ is insensitive to erosion rate when $E^*$ approaches or exceeds one. This prediction is consistent with observations that when erosion rates are high, relief becomes limited by a critical slope angle, set by the material properties of the underlying bedrock (e.g., Binnie et al., 2007; DiBiase et al., 2012). A combination of high $E^*$ and $R^*$ values indicates a landscape with steep, planar hillslopes and frequent landsliding whereas low values suggest more convex hillslopes with lower overall relief (Roering et al., 2007).

For landscapes in topographic steady state with uniform erosion rates, values of $E^*$ and $R^*$ will plot on the steady state curve described by:

$$R^* = \frac{1}{E^*}\left(\sqrt{1+(E^*)^2} - \ln\left(\frac{1}{2}\left(1+\sqrt{1+(E^*)^2}\right)\right) - 1\right).$$ (5)

Here, we define steady state using the formulation of Mudd and Furbish (2004) which considers a hillslope to be in steady state if it retains a constant topographic form with regard to its local base-level, the channel at its base. Steady state hillslopes which experience spatially uniform erosion rates will plot on a single point on the curve (Roering et al., 2007), whereas landscapes experiencing an erosion gradient will plot at many points along this curve, as demonstrated by Hurst et al. (2012). These non-dimensional landscape properties have utility beyond steady state landscapes. Hurst et al. (2013a) used this formulation to distinguish between growing and decaying parts of a landscape by identifying hysteresis in $E^*R^*$ space. Sweeney et al. (2015) has applied similar techniques to analogue landscape evolution models to demonstrate that the efficiency of hillslope sediment transport controls drainage density. These cases of differing landscape properties and histories highlight the power of using topography and $E^*R^*$ analysis to interpret landscape evolution.

The application of such a framework to real data is limited by the challenge of applying a one-dimensional model of hillslope evolution to two-dimensional topographic data. Attempts to apply

such models typically identify non-convergent portions of the landscape upon which to perform tests either through field surveying planar hillslopes (Rosenbloom and Anderson, 1994), the algorithmic identification of convergent topography (Grieve et al., 2016), manual identification of planar topography from digital elevation models or the exclusion of areas of high convergence from hillslope profiles through a valley extraction algorithm as is employed by Hurst et al. (2012) and in this

study. All such methods are compromises between computational efficiency, reproducibility and the accuracy with which a one-dimensional hillslope profile can be extracted. Consequently the conclusions drawn using this, or any other, application of one-dimensional to two-dimensional data must be considered within the context of their potential errors.

## 3 Hilltop patches

The extraction of signals from high resolution topographic data can often require smoothing of raw data to filter out both topographic and artificial noise (Lashermes et al., 2007; Roering et al., 2010; Sofia et al., 2013). This smoothing can be performed either by processing the raw DEM before any analysis is performed (e.g., Roering et al., 2010), or by smoothing the output data (e.g., Tucker et al., 2001; Tarolli and Dalla Fontana, 2009). In order to understand landscape properties at a hillslope

scale it is often desirable to perform local smoothing to group individual DEM pixels into collections of pixels that correspond to individual hilltops and their connected hillslopes.

This was performed by Hurst et al. (2012) through a process of vectorizing hilltops, then splitting the vectors by a threshold length and discarding all split segments shorter than an arbitrary length of 50 meters. The final split vectors are then converted back into rasters, to create a network of hilltop

patches of a defined minimum length. These patches are typically 2 pixels wide, spanning both sides of a drainage divide. This technique is challenging to reproduce, as it relies upon several user defined parameters and a subjective assessment of which vector segments to discard.

### 3.1 Automated generation of hilltop patches

Connected components analysis is a technique typically used in computer vision to label contiguous

pixels in raster images (e.g. Rosenfeld and Pfaltz, 1966; Samet, 1981; Lumia et al., 1983; Dillencourt et al., 1992; Suzuki et al., 2003; He et al., 2013). Here, we implement a computationally efficient connected components algorithm developed by He et al. (2008) to generate contiguous hilltop patches, resulting in a network of hilltop patches, each coded with a unique ID number (Figure 2). Finally, in order to allow better replication of the original concepts used in Hurst et al. (2012), a minimum

patch area can be supplied, which is used to remove any hilltop patches which are smaller than this user defined threshold.

This hilltop patch identification method is very efficient and has been demonstrated to operate effectively on large, complex images (He et al., 2008) without an impact on performance. This technique has utility beyond $E^*R^*$ calculations, as it can be used in any work where discrete patches of hilltop need to be identified (e.g., Perron et al., 2009) or where individual hillslopes must be analyzed using topographic data.

## 4 Generating topographic data

### 4.1 Extraction of a channel network

A key component of most topographic analysis is the delineation of a channel network, without which many topographic parameters cannot be estimated. Channel networks can be extracted either by using a process based method which uses the stream power model to identify the point in a landscape where fluvial processes begin to dominate over hillslope processes (Clubb et al., 2014) or by using a geomorphometric method which identifies channels using curvature thresholds (Passalacqua et al., 2010; Orlandini et al., 2011; Pelletier, 2013).

In order for the $E^*R^*$ data to capture the true range of erosion rates and reliefs inherent in a landscape, it is important to define a channel network which correctly identifies the hillslope-fluvial transition, including the delineation of colluvial channels which are often challenging to identify using non-geomorphometric methods (Pelletier, 2013). Here we follow Pelletier (2013) and apply a Wiener filter (Wiener, 1949) to remove noise from the raw topographic data. Subsequently, channelized portions of the drainage network are identified based on a tangential curvature threshold (e.g., Pelletier, 2013). The appropriate curvature threshold is identified from the properties of its quantile-quantile plot (e.g., Lashermes et al., 2007; Passalacqua et al., 2010). These channelized patches of the landscape are combined by performing a connected components analysis (He et al., 2008) which merges discreet patches of channel into a contiguous channel network. Following methods outlined in Grieve et al. (2016) floodplain masks are also created and combined with this channel network, which separates the landscape into two domains; hillslopes and channels. This has the effect of terminating hillslope traces when they reach a hollow or enter the floodplain, ensuring that the trace properties only reflect the hillslope domain and the $E^*R^*$ measurements are not contaminated by sampling parts of the landscape which the nondimensional framework does not apply to.

If the channel network is incorrectly defined, some fluvial erosion could impact the correct measurement of $E^*R^*$ values. However, due to the number of individual measurements per landscape ($> 160000$ in each case) and the small number of points on a landscape where such erroneous measurements could occur, such measurements will have little impact on landscape scale trends, particularly when spatial averaging is applied.

## 4.2 Extraction of topographic parameters

All of the key measurements required to generate $E^*R^*$ data can be extracted from high resolution topography (Roering et al., 2007). Calculation of $E^*$ using Equation 2 requires hillslope length and hilltop curvature, and calculation of $R^*$ using Equation 4 requires the relief and hillslope length to be measured from high resolution topography.

Grieve et al. (2016) measured hillslope length by generating overland flow paths running from hilltop to channel pixels for every hilltop in a DEM, thereby generating a diverse range of measurements shown to characterize the range of hillslope properties inherent within a landscape. From these traces, each hilltop's local relief is also measured by taking the difference between the elevation at the start (hilltop) and end (channel) of each trace. Finally, the hilltop curvature for each hilltop pixel is extracted following Hurst et al. (2012) whose techniques demonstrated that hilltop curvature scales linearly with erosion rate below hilltop gradients of 0.4. Correspondingly, we also sample the hilltop gradient ($S_{HT}$) at the start of each trace, to allow data to later be filtered by this value. By using the methods outlined by Grieve et al. (2016) we can generate a four-tuple of information for each hilltop pixel in the landscape containing $(L_H, R, C_{HT}, S_{HT})$.

## 4.3 Smoothing topographic parameters

In previous studies that generate $E^*R^*$ data, some form of smoothing has been employed to extract meaningful trends from the inherently noisy topographic data. Roering et al. (2007) hand selected basins with uniform morphologies and minimal anthropogenic disturbance to measure topographic parameters from, effectively removing the majority of noise in the landscape and producing a small number of data points considered to be characteristic of their 2 steady state landscapes.

Hurst et al. (2012) used semi-automated methods to extract the required topographic parameters, and averaged the resulting data spatially over hilltop segments of a defined minimum length. Hurst et al. (2013a) utilized the same methodology, but further averaged the data by grouping segments into bins defined by their distance along the Dragon's Back Pressure Ridge, to explore the topographic expression of a transient uplift signal along the ridge. As these techniques do not self select idealized hillslopes or basins as in Roering et al. (2007) some filtering of the raw data was required (see Section 5.1). These latter methods allow $E^*R^*$ data to be used to interrogate transient landscapes, increasing the power of the method and providing a vital tool in the topographic analysis of landscapes.

Here, we extract topographic parameters from raw topographic data and smooth the resulting measurements, in accordance with previous authors' methods, firstly performing spatial averaging at a basin scale. The basins that are used to average the topographic parameters can be defined in an automated manner to produce an average value over all basins of a given stream order, or a more user defined approach can be undertaken to select basins manually, in order to more closely replicate the work of Roering et al. (2007). Secondly the parameters can be averaged at a hillslope scale by

using the discrete hilltop patches generated using the technique outlined in Section 3.1. The data
     are filtered using the same constraints outlined in Hurst et al. (2012), removing hilltops with a $S_{HT}$
     > 0.4 or a patch size < 50 meters, with the additional filtering of hillslope length and relief values
     below a user defined threshold, typically 2-5 meters for each parameter; this ensures that hilltops
     sampled are true hilltops and are not interfluves sitting adjacent to a basin outlet, which will not
conform to models of hillslope sediment transport. The data are also returned to the user filtered,
     but not averaged, allowing users to explore the raw data to ensure that the smoothed data are a good
     reflection of the overall trends inherent in a landscape. Example basins and hilltop patches, used in
     the smoothing routines and the hillslope traces which produce the topographic measurements are
     displayed in Figure 2.

## 5   Processing the topographic data

     Once the topographic data have been extracted,they are filtered to ensure that only data which con-
     form to the nondimensonal framework described by Roering et al. (2007) are used in any further
     analysis.

     ### 5.1   Filtering

The key filtering process which must be performed is the removal of any data points which have an
     $S_{HT}$ above 0.4. This is the threshold gradient beyond which sediment flux no longer scales linearly
     with slope and thus hilltop curvature does not reflect erosion rate, for $K$ values representative of
     published values for our fieldsites (Roering et al., 1999, 2007; Matmon et al., 2003; Hurst et al.,
     2012). Therefore data points with gradients above this value cannot be used in Equation 2 as a proxy
for erosion rate. Across all of the datasets, gradients which exceed 0.4 are removed from further
     analysis. In the case of the two spatially averaged datasets individual hilltop pixels which exceed
     this threshold gradient within a patch or basin, are removed from the averaging process for each
     measurement ensuring that no invalid data contribute to the final calculations. To ensure the validity
     of each basin average measurement, a count of the valid pixels contained within each basin following
gradient filtering is performed and any basins with fewer valid measurements than a user defined
     threshold can be removed from the analysis. This threshold is typically equal to the minimum patch
     size used in Section 3.1 as this provides consistency between measurements.

     ### 5.2   Log binning

     One method of non-spatial averaging of geomorphic data used effectively to generate slope-area
plots (e.g., Tarolli and Dalla Fontana, 2009) is log-binning. Such a method provides an opportunity
     to interrogate the data at a landscape scale while still removing the noise inherent in topographic
     measurements. Each $E^*R^*$ pair is placed into evenly spaced bins in base 10 logarithmic space. The

bin spacing is a function of the number of bins specified by the user and the range of $E^*$ values within the dataset and its impact on interpretation of the data is considered in Section 6.2.3. To ensure that a valid number of data points make up each bin, a minimum bin size can also be specified by the user, this value will depend on the size and nature of the dataset.

This type of averaging will work best in landscapes where an erosion gradient is expected, as it will produce a range of $E^*R^*$ values across the domain, as can be seen in Hurst et al. (2012). In presumed steady state locations such as Gabilan Mesa most of the data are expected to cluster around a single point (Roering et al., 2007), and so imposing evenly spaced bins in log space onto such data may construct an artificial trend. It is therefore recommended to consider the raw data in conjunction with the binned data to ensure that the trends in the data are valid.

### 5.3 Visualizing data

The software allows the user to plot any combination of the $E^*R^*$ datasets, facilitating the rapid generation of basin and landscape average data following Roering et al. (2007), hilltop-averaged and log-binned data following Hurst et al. (2012, 2013a) and raw data which has previously not been available. It is also possible to interrogate the raw measurements as a density plot, which more accurately conveys the trends in the raw data as in large landscapes many measurements share the same location in $E^*R^*$ space. By allowing simple inter-comparisons between plotting methods it becomes trivial to assess the most suitable data visualization techniques for a specific landscape.

## 6 Results and discussion

By using data from previous studies which utilize $E^*R^*$ analysis it is possible to assess the ability of this software to reproduce existing results in addition to understanding how the varying techniques for smoothing the data, discussed in Section 4.3, can impact on the interpretation of the processes operating on a landscape. Four landscapes in the continental USA have been selected to evaluate the software, the Oregon Coast Range and Gabilan Mesa, used by Roering et al. (2007), Cascade Ridge, used by Hurst et al. (2012) and the Coweeta Hydrologic Laboratory (Figure 1). High resolution LiDAR data are available from the National Center for Airborne Laser Mapping (NCALM) for each site and each site's point cloud data have been gridded to 1 meter resolution DEMs following Kim et al. (2006) and accuracy information for each point cloud can be found in Appendix A.

### 6.1 Reproducing previous work

#### 6.1.1 Oregon Coast Range and Gabilan Mesa

The Oregon Coast Range in Oregon, USA is a steeply incised upland landscape with dense forest cover and a humid climate (Roering et al., 1999), leading to frequent debris flows, which initiate in colluvial hollows (Stock and Dietrich, 2003). The forests of the Oregon Coast Range are dominated

by hardwoods, such as Oregon Maple (*Acer macrophyllum*), and coniferous forest such as Douglas Fir (*Pseudotsuga menziesii*) (Schmidt et al., 2001). Extensive work has been carried out to estimate the uplift rate of the range using marine terrace data (Kelsey et al., 1996), and these estimates of uplift rate correspond to erosion rates measured using cosmogenic radionuclides (e.g., Heimsath et al., 2001). This correspondence between uplift and erosion rate has been used to infer that the Oregon Coast Range is in steady state (e.g., Reneau and Dietrich, 1991; Roering et al., 2007).

Gabilan Mesa in California, USA is part of the Central Coast Ranges and has a semiarid Mediterranean climate with higher vegetation densities on northern slopes due to microclimatic variations (Dohrenwend, 1978). The vegetation of Gabilan Mesa is characterized by a combinations of oak savannah containing Blue Oak (*Quercus douglasii*) and chaparral shrubland containing Chamise (*Adenostoma fasciculatum*) (Shreve, 1927). The landscape is very smooth with a regular spacing of tributaries and valleys (Dohrenwend, 1978, 1979) and gentle transitions between hillslopes and channels, suggesting that diffusive processes dominate the transport of sediment on hillslopes (Roering et al., 2007). Hilltop curvature shows little variance across the landscape and in conjunction with the regularity of valley spacing, this suggests that the landscape is in approximate topographic steady state (Roering et al., 2007; Perron et al., 2009).

Roering et al. (2007) estimated the topographic parameters $L_H$, $R$ and $C_{HT}$ for the Oregon Coast Range and Gabilan Mesa fieldsites. The characteristic hillslope length for each landscape was estimated by identifying the inflection point in a spline curve fitted though a plot of local slope against drainage area. This inflection point is considered to correspond to the transition between the hillslope and channel domain in a landscape (Montgomery and Foufoula-Georgiou, 1993; Hancock and Evans, 2006; Tarolli and Dalla Fontana, 2009; Tarolli, 2014; Tseng et al., 2015).

Roering et al. (2007) estimated mean relief by calculating the mean of the differences between the maximum and minimum elevation within a kernel of radius equal to the characteristic hillslope length for each point on the landscape. Hilltop curvature was sampled from manually defined hilltops with a gradient below $0.05S_c$ and averaged across each landscape. The critical gradient was calculated for the Oregon Coast Range to be 1.2 by Roering et al. (1999) and Roering et al. (2007) assumed that this value is also correct for Gabilan Mesa.

The data from Gabilan Mesa (Figure 3A) reveal many hilltop patches which correspond closely to the predicted $E^*R^*$ values from Roering et al. (2007). The data are predominantly clustered around a single point, showing strong agreement with observations that the landscape is in approximate steady state. However the majority of the basin average data points and a considerable amount of the hilltop patch data plots below the steady state curve, which could be interpreted as evidence for topographic decay. However the uniform hilltop curvatures and valley spacing, coupled with measurements of long term erosion rates suggest that this landscape is not undergoing topographic decay (Roering et al., 2007; Perron et al., 2009). An alternative explanation for the data falling below the steady state curve is that an $S_c$ value of 1.2 is too large for this landscape. Grieve et al. (2016) used similar

topographic parameters to estimate the critical gradient for this landscape as 0.8. By replotting these data using this revised $S_c$, the data plot more closely to the steady state curve (Figure 3B).

The Oregon Coast Range data are more tightly constrained than the Gabilan Mesa data (Figure 4A), and have a similar range of $R^*$ values. However, as is the case for Gabilan Mesa, the majority of the data plot below the steady state curve. This can be interpreted as evidence for topographic decay, however due to the preponderance of evidence supporting a steady state hypothesis for this landscape (e.g., Reneau and Dietrich, 1991; Roering et al., 2007), it is also possible that a critical

gradient of 1.2 is too large in this location. By using the $S_c$ value of 0.79 constrained by Grieve et al. (2016), the data move closer to the steady state curve (Figure 4B). Using this average $S_c$ value several $R^*$ measurements exceed 1. This indicates that these hillslopes are too steep to sustain soil mantle in this landscape, which corresponds to field observations of the Oregon Coast Range, where frequent shallow landsliding is reported (e.g., Benda and Dunne, 1997; Montgomery et al., 1998)

and where periodic wildfires expose large (tens of $m^2$) patches of bedrock (Jackson and Roering, 2009).

As acknowledged by Roering et al. (2007) extracting the relief from a moving window fails to capture the complete range of relief values in a landscape, resulting in an average value which dampens the true signal, reducing $R$ in high relief landscapes such as the Oregon Coast Range. Our method

of measuring relief of individual hillslope traces circumvents this problem.

The majority of the data points in Figures 3 and 4 have larger $E^*$ values than those from Roering et al. (2007). Grieve et al. (2016) showed that estimating $L_H$ using slope-area plots systematically underestimates $L_H$ by as much as an order of magnitude in some landscapes. Such an underestimate would reduce the $E^*$ value for a landscape and explains the systematic differences between this

study and the results of Roering et al. (2007). The larger range of hilltop patch data highlights the range of $E^* R^*$ values inherent in even a uniform landscape which is in approximate topographic steady state.

### 6.1.2    Cascade Ridge

Cascade Ridge is a section of the Northern Sierra Nevada in California, USA. The landscape is pre-

dominantly forested and the climate is semi-arid (Hurst et al., 2012). The characteristic topographic form of this landscape is a smooth, low relief relict surface which is heavily incised, creating steep canyons with an irregular spacing. The plateau surface is vegetated with oak forest including California black oak (*Quercus kelloggii*) and canyon live oak (*Quercus chrysolepis*) and pine forest containing ponderosa pine (*Pinus ponderosa*), Douglas fir (*Pseudotsuga menziesii*) and sugar pine

(*Pinus lambertiana*), whereas the canyon is dominated by chaparal vegetation such as manzanita (*Arctostaphylos spp*) (Gabet et al., 2015; Milodowski et al., 2015a). These contrasting landscape morphologies have been shown to be eroding at different rates, with the plateau surfaces eroding an order of magnitude more slowly than the canyons (Riebe et al., 2000; Hurst et al., 2012). This

produces a complex landscape exhibiting a range of erosion rates influenced by climate and tectonic signals, which is not in topographic steady state (Riebe et al., 2000; Stock et al., 2004; Hurst et al., 2012; Gabet et al., 2015).

Cascade ridge is a more morphologically complex landscape than the Oregon Coast Range or Gabilan Mesa, correspondingly, the $E^*R^*$ data for this landscape are predicted to plot along the steady state curve at a broad range of $E^*$ values, as was demonstrated by Hurst et al. (2012). Using an $S_c$ value of 0.8, as proposed by Hurst et al. (2012), produces data spanning a much wider portion of $E^*R^*$ space than the data for the steady state landscapes of Gabilan Mesa and the Oregon Coast Range (Figure 5A). The binned hilltop patch data show good agreement with the data from Hurst et al. (2012), spanning a similar range of $E^*$ values with the steady state curve falling within the standard error of each bin. This supports observations of a range of erosion rates and landscape morphologies and highlights the utility of this method in gaining a first order approximation of the tectonic and erosional setting of a landscape where no field data are available.

At the Cascade Ridge site, Grieve et al. (2016) estimated $S_c$ to be 0.72, calculated from topographic parameters. Using this value there is little change in the trends in the data (Figure 5B), most of the points now fall above the line and at high values of $E^*$, and more data points have $R^*$ values in excess of 1. These high $R^*$ values are consistent with field observations of this transient landscape wherein rapid valley downcutting may decouple hillslopes from the channel network (Milodowski et al., 2015b) and drive shallow landsliding. In a complex landscape such as Cascade Ridge, which is known to have a broad range of erosion rates and hillslope morphologies, a landscape average $S_c$ value will regress towards the mean. Consequently, as more of the landscape is covered by the low gradient plateau than the steeper canyons, the $S_c$ value of 0.72 does not reflect the parts of the landscape with larger $E^*R^*$ values, which may fall closer to the value of 0.8 used by Hurst et al. (2012).

### 6.1.3 Coweeta

The Coweeta Hydrologic Laboratory is in the Southern Appalachian Mountains in North Carolina, USA and is a densely vegetated landscape which exhibits classic ridge and hollow topography (Hales et al., 2012). Such topography produces many source areas for shallow landslides in colluvial hollows, which are triggered by high intensity storms connected to hurricanes (Swift Jr. et al., 1988). The vegetation at Coweeta is a mix of shrubs, such as *Rhododendron maxima*, and Northern Hardwood forest, the distribution of which is controlled by wildfires, which in many cases are managed through human intervention (Hales et al., 2009). It is debated whether the Southern Appalachians are in topographic steady state, as there is little tectonic activity, yet there is a large amount of relief preserved across the range (Baldwin et al., 2003; Matmon et al., 2003; Gallen et al., 2011, 2013).

The Southern Appalachian Mountains have never previously been investigated using $E^*R^*$ methods and so can be used to evaluate the technique's ability to interrogate a complex landscape and

assist in the interpretation of topographic signals. Figure 6 outlines the range of methods which can

be used to interpret $E^*R^*$ data. As in Sections 6.1.1 and 6.1.2 the critical gradient used is taken from

Grieve et al. (2016). The raw data in Figure 6A shows the range of reliefs observed in the South-

ern Appalachians. The landscape median $E^*R^*$ value falls within the zone of maximum probability

density, this highlights the level of noise inherent in high resolution topographic data when interro-

gating them in $E^*R^*$ space, outlining the requirement to smooth or bin the data in order to extract

meaningful information from them.

Comparing the data in Figure 6B and 6C to data for steady state landscapes such as Gabilan

Mesa or the Oregon Coast Range, they show similar levels of clustering, with the location of the

cluster of patch and basin average values corresponding with the Oregon Coast Range data (Figure

4). This corresponds well to field observations of hillslope morphology in these two locations, with

planar hillslopes and frequent shallow landsliding reported (Benda and Dunne, 1997; Montgomery

et al., 1998; Roering et al., 1999) and this clustering suggests that there is less spatial variation in

erosion rate in Coweeta than in Cascade Ridge, an assertion supported by measured erosion rates

from both locations (e.g., Riebe et al., 2000; Matmon et al., 2003; Hales et al., 2012; Hurst et al.,

2012). Figure 6D shows the binned data for Coweeta and highlights the smaller range of $E^*$ values

for this landscape when compared to Cascade Ridge. It also draws attention to the need to analyze

$E^*R^*$ data using numerous methods to avoid an incorrect interpretation, as discussed in Section 5.2.

The Coweeta $E^*R^*$ data clusters around a point on the steady state curve and it could be con-

cluded that this landscape is in approximate steady state. However, the value of $S_c$ used in Figure

6 is significantly smaller than any previously published $S_c$ value. Field observations of Coweeta

reveal that many channels are alluviated and such deposition at the base of hillslopes will alter the

mean properties of a hillslope, and move its idealized profile away from the model hillslopes defined

by Roering et al. (2007). As a valley fills with sediment, the hillslope relief will be reduced more

rapidly than other hillslope properties, due to the difference between rates of hillslope and channel

response to forcing (Hurst et al., 2012). Such a reduction in relief will reduce $R^*$, resulting in a

reduced best fit $S_c$ value. Such an alteration of mean hillslope properties could explain the consid-

erable underestimation of the critical gradient when it is constrained through hillslope length-relief

relationships.

The Oregon Coast Range, a broadly similar landscape to Coweeta, based on the range of $E^*$ val-

ues, general landscape morphology and observations of sediment transport processes, has a critical

gradient of 0.79 (Grieve et al., 2016). This value is similar to the $S_c$ of many other landscapes (DiB-

iase et al., 2010; Hurst et al., 2012; Grieve et al., 2016) and as such we use this value to explore the

patterns of $E^*R^*$ in Coweeta when a larger critical gradient, which more closely resembles predicted

values for other landscapes is employed. In such a case the majority of the data plot below the steady

state curve (Figure 7). Hurst et al. (2013a) observed $E^*R^*$ data plotting below the steady state curve,

along the Dragons Back Pressure Ridge, where these sections of the landscape are understood to be

topographically decaying following a pulse of uplift. If this $S_c$ is correct it could lend support to the idea of a Miocene rejuvenation of topography in the Southern Appalachians (Gallen et al., 2013) followed by a period of gradual topographic decay into the present. However, the nature of sediment transport in Coweeta may not be best constrained using Equation 1 as modeling work performed by Mudd (2016) suggests that a deviation of this magnitude from the steady state curve indicates that a landscape is not undergoing pure nonlinear sediment flux.

## 6.2 Sensitivity analysis of averaging methods

Several of the techniques utilized to average the raw $E^*R^*$ data have free parameters, the selection of which can influence the final results. In the following section we explore the influence that averaging technique, minimum patch and basin area, basin stream order and binning parameters can have on the interpretation of $E^*R^*$ data.

### 6.2.1 Averaging methods

As outlined in Section 4.3, the topographic parameters, $L_H, R, C_{HT}$, and $S_{HT}$, must be smoothed in order to extract meaningful trends from the inherently noisy signal. The main technique for performing this smoothing is to spatially average the data over either hilltop patches or drainage basins. These averages can be computed as either the mean or the median of each spatial area. Figure 8 presents a comparison between hilltop patch data computed using means and medians for the Oregon Coast Range, showing little change between the measurements using the two techniques. Because there is little difference between the two methods, we use median values throughout this paper, as this ensures that any extreme values will have a lesser impact on landscape scale metrics.

### 6.2.2 Spatial averaging parameters

The hilltop patch identification process described in Section 3.1 requires one user defined parameter, the minimum patch area. This value is used to remove any small patches from the analysis and is included to ensure that patches conform to geomorphologically significant hillslopes, and not small patches of hilltop that are not representative of the hillslope as a whole. By varying the size of the minimum patch area from 0 through to 500 pixels it is possible to observe how this parameter can impact the interpretation of $E^*R^*$ data (Figure 9A). As the threshold is increased, fewer patches are considered valid and the density of the data are reduced, having the effect of removing many of the outlying data points. This reinforces the need to set a minimum size for a hilltop patch to ensure that a small number of measurements do not have too large an impact on the interpretation of the data.

The technique in Section 3.1 has no method to limit the maximum size of the hilltop patches, as the aim is to find spatially contiguous zones of hilltop and artificially breaking these patches may result in oversampling some sections of a landscape. Large patches make up a very small proportion

of the total population of patches and correspondingly do not have a large impact on the overall trends in an individual dataset.

The stream order of the basin used to generate basin average values will also have an influence on the interpretation of the results. Grieve et al. (2016) used second order basins to generate basin average topographic parameters as this order generated a large number of basins which all had a large enough area to generate numerous data points per basin effectively sampling as much of the landscape as possible. Figure 9B shows the effect of increasing the stream order of the basins used in Coweeta from first to fourth order. As each increasing order basin can be considered a set containing the previous order basins, the basin average points all plot in very similar locations in $E^*R^*$ space, suggesting that increasing basin order may be a useful method of smoothing basin average data 480 in noisy landscapes. However, this comes with the limitation that as the basin order increases, the number of basins in a landscape decreases, resulting in fewer data points representing larger spatial areas and the possible homogenization of topographic signals occurring at spatial scales smaller than the average basin area.

The number of valid data points contained within a basin used to generate an average value is 485 another free parameter that the user must set. As with the hilltop patch area, selecting a sensible value is important to ensure that each basin average data point corresponds to the basin as a whole and not just a spatial subset. As the threshold is increased, outlying basins are removed (Figure 9C), indicating that many outlying data points are generated by a small number of irregular hillslopes in otherwise typical basins. However, if the threshold is too large, too many basins will be excluded. 490 In order to ensure consistency between spatial averaging techniques it is recommended that the minimum number of pixels in a basin be kept equal with the minimum patch area.

### 6.2.3 Log bin parameters

When computing logarithmically spaced bins there are two free parameters, number of bins, equivalent to the bin width and the minimum number of data points which must fall within a bin for the 495 binned point to be valid. Figure 10A highlights the influence of changing the number of bins on the interpretation of the Cascade Ridge data. If the number of bins are too low, it becomes difficult to identify a trend in the data as the nature of a landscape can vary considerably across large ranges of $E^*$ and by homogenizing these measurements a transient signal can be lost.

However, as the number of bins is increased, fewer values are placed into each bin, meaning that 500 if there is a single value which is significantly different to the rest of the values in the bin, it can vastly alter the result. It is also the case that as the number of bins increases, the chance of a bin being removed for having to few data points increases, which will be particularly apparent at low and high $E^*$ values, where the data are sparser. We have found that using 20 bins reaches a good compromise between data density and data smoothing, and corresponds well with the 21 bins used 505 by Hurst et al. (2012), where no filtering was performed based on bin size.

The minimum size of each bin can also have an impact on the final interpretation of the data. If no threshold is applied, some bins can contain a single value, while others can contain hundreds of values which makes interpreting the data difficult as one cannot be sure of the robustness of each binned value. If the threshold is placed too high, then valid data will not be included in the final analysis and the interpretation of a landscape's evolution could be incorrect. Figure 10B highlights this issue using data from Cascade Ridge at a range of bin size thresholds, identified as percentages of the total dataset size. We have found that using a minimum bin size of 1-5% of the total dataset ensures a good binning result.

### 6.3 Constraining $S_c$

Landscapes which are in topographic steady state should plot at a single location on the curve described by Equation 5. In principle this would mean that an erosion gradient would be required in order to constrain $S_c$, by fitting the data to the steady state curve. However, as observed in Figures 3 and 4 even in idealized steady state landscapes, there is still considerable variability in the $E^*R^*$ data. This variability is consistent with patterns of dynamic reorganization of low order drainage basins within models of steady state landscapes performed by Reinhardt and Ellis (2015). Therefore it becomes possible to estimate the critical gradient of the nonlinear sediment flux law (Equation 1) for a landscape without a strong erosion gradient, using $E^*R^*$ data.

As with previous analyses, the raw data must be spatially averaged in order to reduce the level of noise present in $E^*R^*$ space before an estimate of $S_c$ can be made. The optimal value of $S_c$ is estimated using a nonlinear least squares method (Jones et al., 2001) which computes the sum of the square of the deviation between each measured $E^*R^*$ value and the value predicted by Equation 5. This calculation is performed for a range of critical gradients until the $S_c$ with the lowest corresponding deviation from the steady state curve is found.

The accuracy of this optimized $S_c$ value is constrained through bootstrapping the optimization procedure. The data are sampled with replacement to generate $100\,000$ datasets, consisting of values randomly drawn from the population of patch or basin average data. For each of these sampled datasets the optimal value of $S_c$ which minimizes the error between the data and the steady state curve is calculated. The final $S_c$ value for each landscape is the mean value of these $100\,000$ iterations, with a 95% confidence interval.

Table 1 contains estimates of the critical gradient generated using both basin and patch average values alongside previously published values for Cascade Ridge, the Oregon Coast Range and Gabilan Mesa. The predicted patch and basin average values for Gabilan Mesa and the Oregon Coast Range are similar to those published by Grieve et al. (2016). This method of estimating the best fit $S_c$ will produce an average value representative of the maximum probability density of $S_c$ values for a landscape. Whereas the method of $S_c$ estimation employed by Roering et al. (1999) can better be considered as the maximum $S_c$ value for a landscape Grieve et al. (2016).

The data for Cascade Ridge shows better agreement with the value used by Hurst et al. (2012), which was also derived using $E^*R^*$ data, than the lower estimate from Grieve et al. (2016). The pair of $S_c$ values calculated for each landscape are very similar, suggesting that in large enough datasets, the constraint of $S_c$ is insensitive to the spatial scale of data averaging. However, the scale of spatial averaging has been demonstrated to have an impact on the interpretation of $E^*R^*$ data and thus care must be taken to select appropriate methods of spatial averaging and data processing in order to ensure that results generated are not simply a function of user defined parameters.

The similarity of the average $S_c$ values obtained using the bootstrapping procedure across three diverse landscapes highlights the presence of a distribution of $E^*R^*$ values existing for each landscape, and the nature of an average $S_c$ measurement. Such a distribution occurs due to local variations in topography, process and material properties and similarities can be drawn between the results presented in Table 1 and other similar studies (DiBiase et al., 2010; Hurst et al., 2012).

The values of $S_c$ constrained using this bootstrapping procedure are similar to those derived from the relationship between hillslope length and relief demonstrated by Grieve et al. (2016), however there is no need to estimate material properties such as the soil and rock density and thus this method provides an independent constraint on $S_c$. However, the computational expense of bootstrapping the $S_c$ fitting calculations from the $E^*R^*$ data is very high, when contrasted with the estimation of $S_c$ using $L_H - R$ relationships presented by Grieve et al. (2016). Additionally, using this bootstrapping method in landscapes which do not plot on the steady state curve in $E^*R^*$ space can yield an incorrect $S_c$ value with a low error estimate. Consequently, we recommend estimating the critical gradient of a landscape using this method and the method outlined in Grieve et al. (2016), when field data are available, in order to best constrain the critical gradient of a landscape. However, careful consideration of the differences between a maximum $S_c$ and a best fit derived average $S_c$ should be undertaken to ensure that a valid geomorphic interpretation of a landscape is employed.

## 7 Conclusions

We present a software package which automates the extraction and processing of high resolution topographic data to generate non-dimensional erosion rate and relief measurements. Topographic data can be averaged at a hilltop scale by generating unique hilltop patches or can be averaged over drainage basins automatically extracted from the channel network. Alongside the raw data, these spatially averaged datasets are shown to reproduce the findings of previous studies. In steady state landscapes such as the Oregon Coast Range and Gabilan Mesa $E^*R^*$ data plot in a cluster around a single point on the steady state curve, supporting the conclusions drawn in previous studies (Roering et al., 2007); and in Cascade Ridge, a transient erosion signal similar to that identified by Hurst et al. (2012) is observed. This technique is also tested on a landscape in the Southern Appalachian mountains with the results suggesting that topography is decaying, supporting models of Miocene

topographic rejuvenation proposed by Gallen et al. (2013). These results, alongside the ability to reproduce previous work emphasizes the value of this software to the geomorphology community as until now, there has been no clear framework within which to produce non-dimensional erosion rate and relief measurements.

The average critical gradient used in Equation 1 is also constrained for three of the studied landscapes, with the values falling within expected ranges. However due to the noise inherent in this form of analysis and the challenges of evaluating the goodness of fit between such noisy data and a model, it is recommended that other methods to constrain $S_c$ using the same raw data are utilized instead. Finally, the influence of free parameters on the final interpretation of the data are explored, providing the user clear guidance on how to select parameters which control the level of smoothing or binning performed on the topographic data. The most significant of which are the minimum patch and basin size thresholds which must be carefully selected to balance smoothing the data with preserving landscape scale trends.

*Author contributions.*

SWDG developed the data analysis and visualization code and performed the data analysis. SWDG, SMM, MDH and DTM wrote the topographic analysis code. SWDG wrote the manuscript with contributions from SMM, MDH and DTM.

*Acknowledgements.* The topographic data used in this paper are freely available from http://www.opentopography.org. All the code used in this analysis is open source and can be downloaded from https://github.com/LSDtopotools/ LSDTT_Hillslope_Analysis, http://github.com/sgrieve/ER_Star/ and http://github.com/sgrieve/ER_Star_Figs/. The data used to generate all the plots are available from http://www.geos.ed.ac.uk/~s0675405/ER_Data/ER_ Data.zip. SWDG is supported by NERC grant NE/J009970/1. SMM is supported by U.S. Army Research Office contract number W911NF-13-1-0478. DTM is supported by NERC grants NE/152830X/1 and J500021/1, in addition to the Harkness Award from the University of Cambridge. This paper is published with the permission of the Executive Director of the British Geological Survey and was supported in part by the Climate and Landscape Change research programme at the BGS. We thank Fiona Clubb for discussions and advice which have shaped the final form of this manuscript. This manuscript has been improved through the detailed reviews provided by Jon Pelletier, an anonymous reviewer and the Associate Editor, Susan Conway.

**Appendix A: Topographic Metadata**

This table provides the accuracy information for the four point clouds used to generate the 1 meter resolution topographic data used in this study. This information is compiled from metadata available from OpenTopography.org.

**Table A1.** LiDAR point cloud metadata.

| Location | Point density (points per $m^2$) | Vertical accuracy (m) | Horizontal accuracy (m) |
|---|---|---|---|
| Oregon Coast Range | 6.55 | $0.07 \pm 0.03$ | 0.06 |
| Gabilan Mesa | 5.56 | $0.20 \pm 0.15$ | 0.11 |
| Cascade Ridge | 9.84 | $0.17 \pm 0.13$ | 0.11 |
| Coweeta | 8.91 | $0.17 \pm 0.13$ | 0.11 |

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

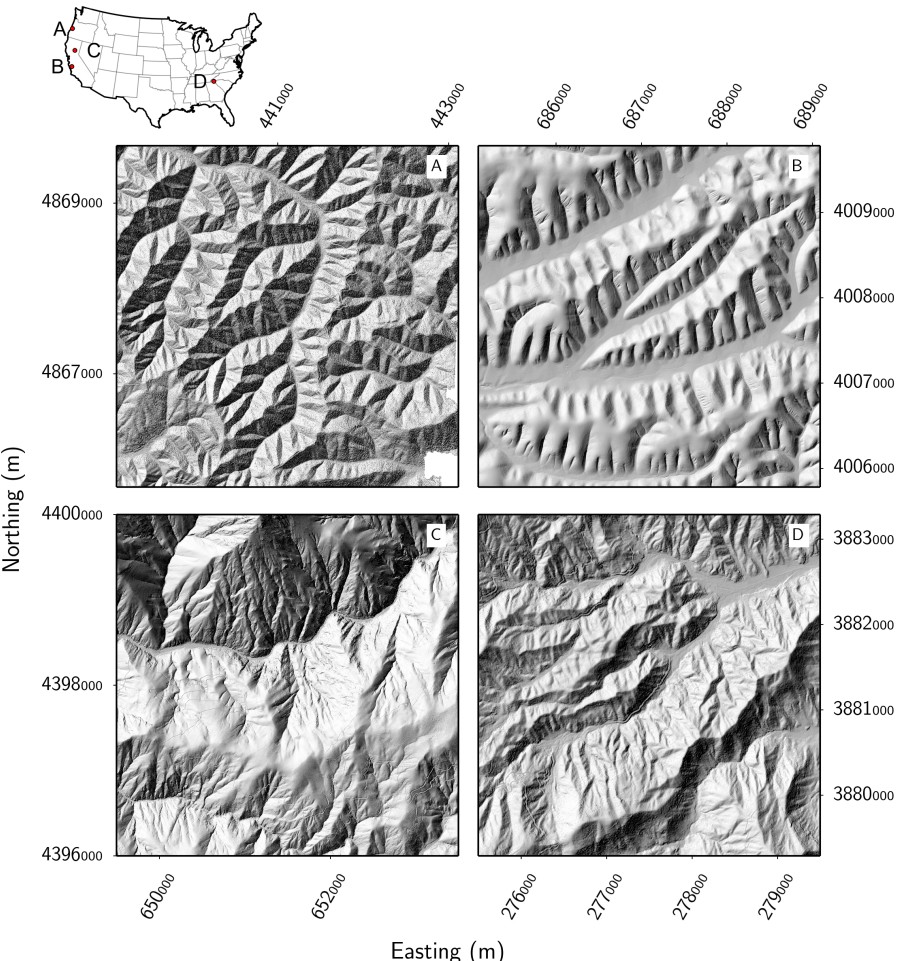

**Figure 1.** Map of the locations of each field site within the continental USA with a shaded relief map of characteristic sections of each location's topography. All coordinates are in UTM. (A) Oregon Coast Range, Oregon, UTM Zone 10N. (B) Gabilan Mesa, California, UTM Zone 10N. (C) Northern Sierra Nevada, California, UTM Zone 10N. (D) Coweeta, North Carolina, UTM Zone 17N.

**Table 1.** Previously published $S_c$ values alongside the values generated from the best fit to the steady state curve for the patch and basin average data.

|  | Roering et al. (2007) | Hurst et al. (2012) | Grieve et al. (2016) | Patch Average[1] | Basin Average[2] |
|---|---|---|---|---|---|
| Oregon Coast Range | $1.2 \pm 0.2$ | — | 0.79 | $0.83 \pm 0.01$ | $0.83 \pm 0.01$ |
| Gabilan Mesa | $1.2 \pm 0.4$ | — | 0.8 | $0.8^{+0.06}_{-0.05}$ | $0.8^{+0.05}_{-0.04}$ |
| Cascade Ridge | — | 0.8 | 0.72 | $0.78 \pm 0.02$ | $0.82 \pm 0.02$ |

[1] Calculated as the value which minimizes the sum of the squared residuals to the steady state line for the patch average data. Error is the 95% confidence interval generated by bootstrapping the calculation $100\,000$ times.

[2] As for [1] but using basin average data.

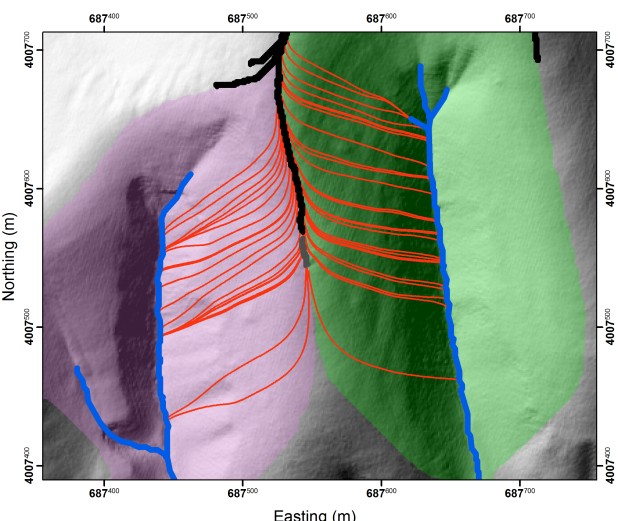

**Figure 2.** Map of a section of Gabilan Mesa, California (UTM Zone 10N), showing the examples of the spatial units used in the analysis of $E^* R^*$ data. Two second order basins, colored green and purple, are bisected by a ridge with two hilltop patches, a large black patch and a smaller grey patch. From these patches representative hillslope traces, outlined in red, travel down the hillslope and terminate at the channel network. Only 10% of the total traces generated for this ridge have been plotted and other surrounding hilltop patches and their associated traces are not displayed, to aid clarity. Coordinates are in meters.

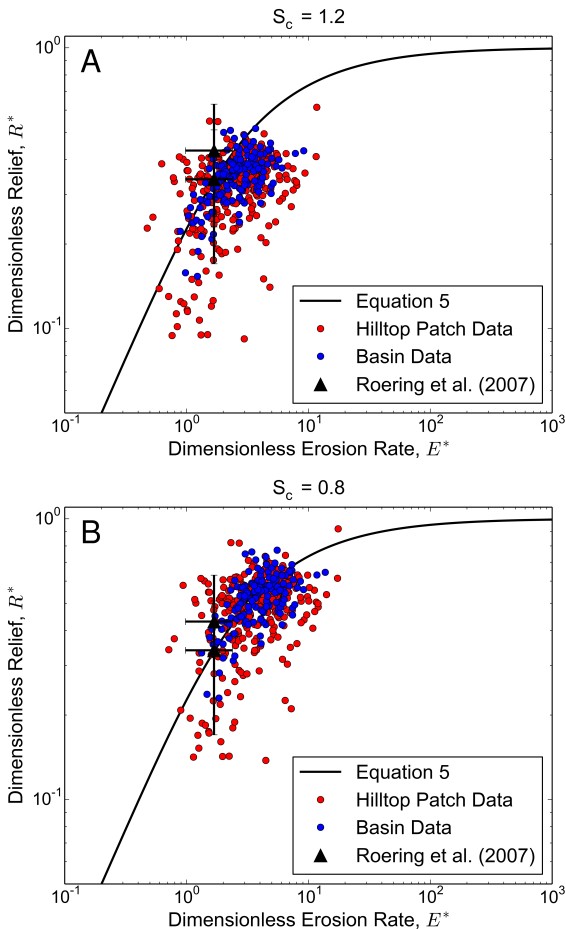

**Figure 3.** Hilltop patch and basin average data for Gabilan Mesa plotted using a critical gradient of 1.2 (A) and 0.8 (B) alongside data from Roering et al. (2007) for the same location. Errorbars are the standard error.

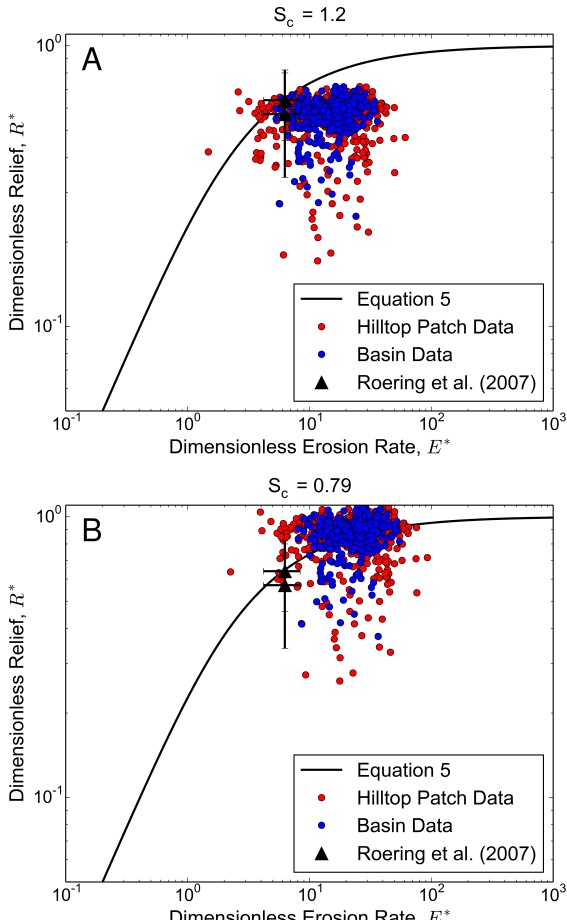

**Figure 4.** Hilltop patch and basin average data for the Oregon Coast Range plotted using a critical gradient of 1.2 (A) and 0.8 (B) alongside data from Roering et al. (2007) for the same location. Errorbars are the standard error.

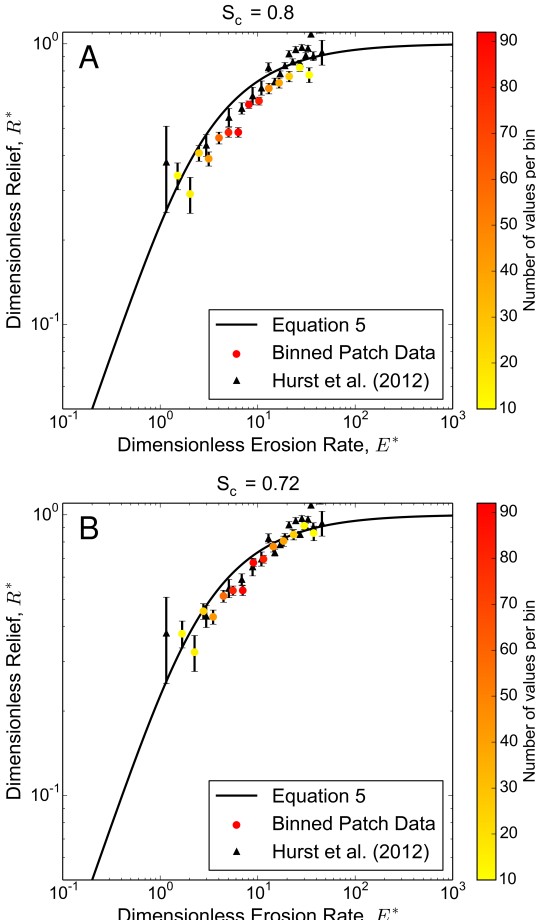

**Figure 5.** Binned hilltop patch data spanning a wide range of $E^*$ values generated using a critical gradient of 0.8 (A) and 0.72 (B) alongside data from Hurst et al. (2012) for the same location. Errorbars are the standard error of the data. Errorbars from Hurst et al. (2012) are generated from the original data.

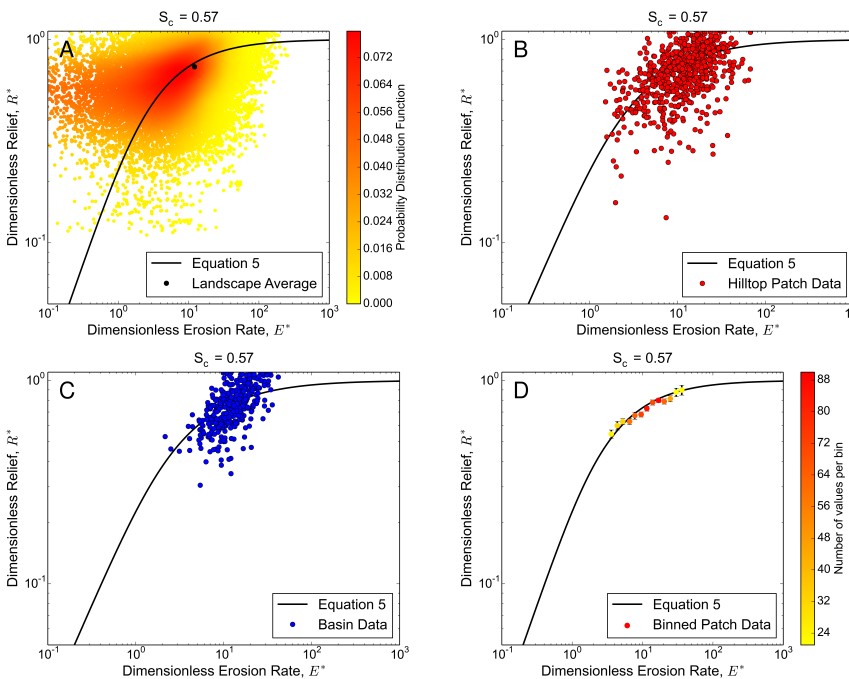

**Figure 6.** Comparison of the different methods which can be used to visualize $E^*R^*$ from Coweeta, using a critical gradient of 0.57. (A) Raw data colored by the density of points in $E^*R^*$ space alongside the landscape average value. Errorbars plot inside the data point. (B) Data averaged over hilltop patches. (C) Data averaged over second order drainage basins. (D) Hilltop patch data placed into logarithmically spaced bins, errorbars are the standard error.

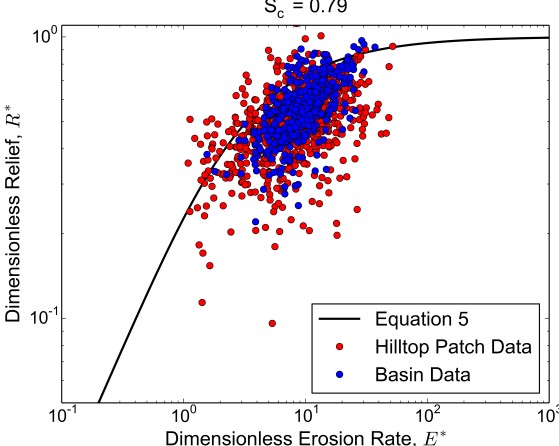

**Figure 7.** Hilltop patch data from Coweeta plotted using the higher $S_c$ value of 0.79, demonstrating that the majority of the hillslopes in this landscape plot below the steady state curve when using a larger critical gradient.

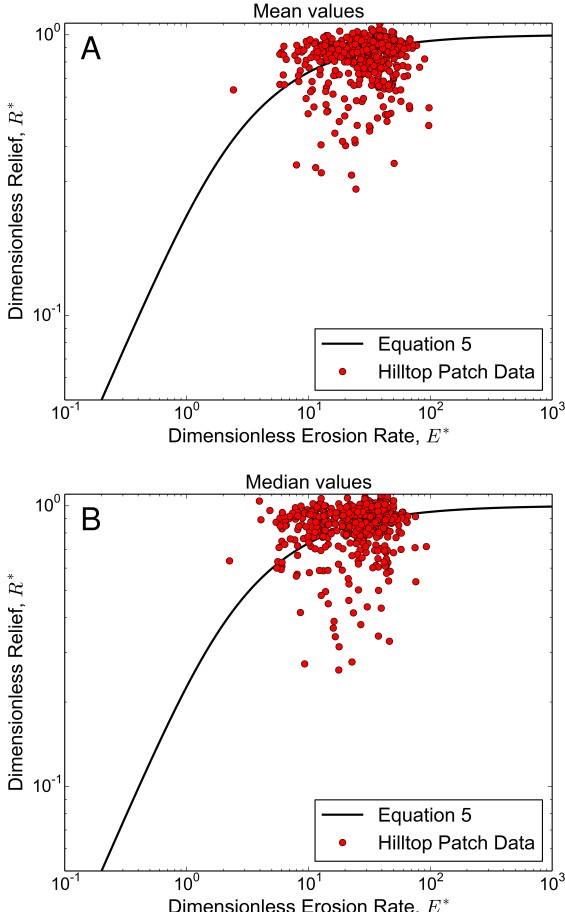

**Figure 8.** Comparison between hilltop patch values generated using a spatial mean (A) and a spatial median (B) for the Oregon Coast Range.

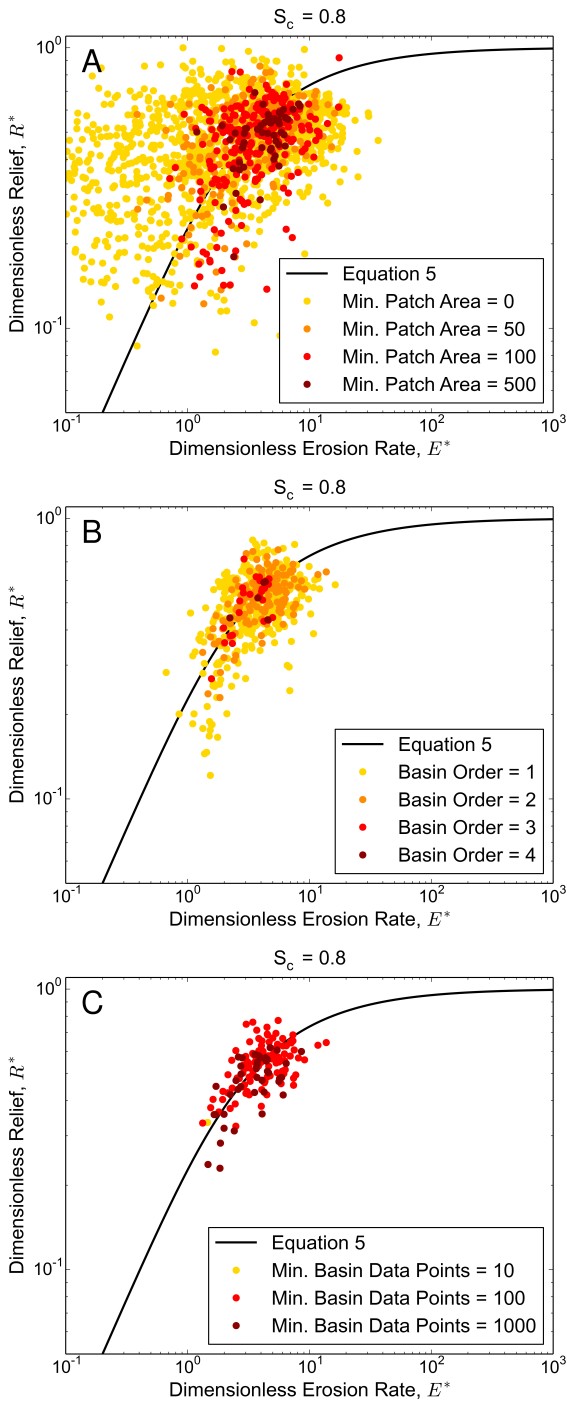

**Figure 9.** Comparison of the influence of changing spatial averaging method on the interpretation of $E^* R^*$ data for Gabilan Mesa. (A) variations in the minimum patch area threshold from 0 (no threshold) to 500 pixels highlighting the reduction in noise when a minimum patch area is applied. (B) increasing the basin steam order, which reduces variance in the data, as bigger basins are sets containing basins of smaller orders, dampening any extreme values. (C) variations in the minimum basin pixels threshold. Outlying basins have very few data points, so are influenced more strongly by single atypical values.

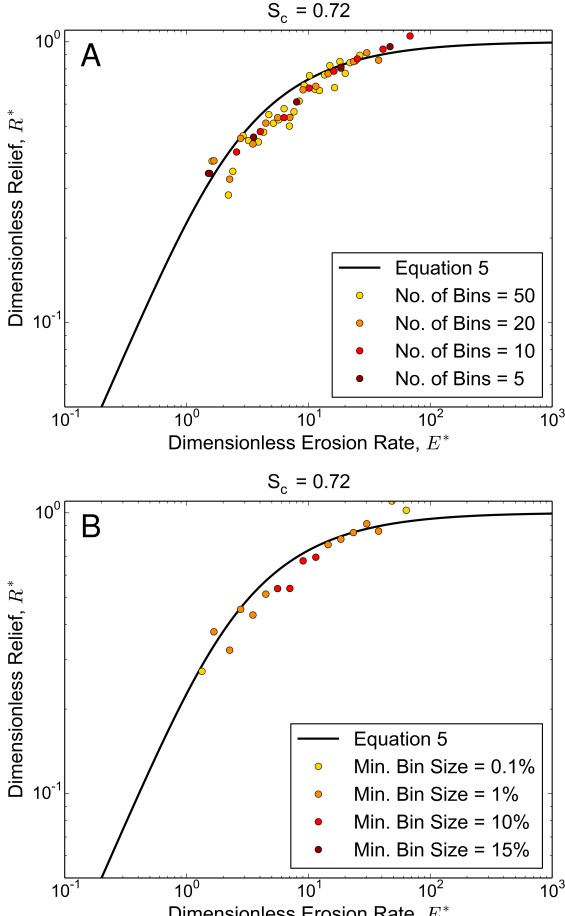

**Figure 10.** Comparison of the influence of binning parameters on the interpretation of $E^* R^*$ data for Cascade Ridge. (A) varying the number of bins used, equivalent to the bin width in $E^*$ space. As the number of bins reduces it becomes harder to identify patterns in the data and as the number of bins increases, the number of data points in each bin reduces, thereby reducing the power of the binning technique. (B) varying the minimum number of data points required in a basin. As this value increases fewer points are preserved, which compresses the range of the data and can obscure the observation of an erosional gradient. Too small a threshold can result in bins containing very few values which do not represent the landscape as a whole.