# Peer review of "A nondimensional framework for exploring the relief structure of landscapes"

_Earth Surface Dynamics, 2015_

## Referee Comment (RC1) · J. Pelletier (Referee) · 8 Feb 2016

Grieve et al. propose a framework for computing the dimensionless relief and erosion rates of ridge-and-valley topography using a combination of valley network extraction and hilltop curvature analysis. They argue that their approach allows one to determine whether or not a landscape is in topographic steady state and to determine the mean Sc value (the maximum gradient of stability) of a landscape.

Overall I think the paper will be an excellent contribution, once some issues are thoroughly considered and/or caveats provided.

1) I am skeptical that the mean value of Sc is 0.79 in the Oregon Coast Range. Roering et al. (1999) demonstrated that many hillslopes in the OCR have gradients in the 0.8 to 1.1 range, and more importantly that the planarity of hillslopes systematically increases

as gradients approach 1.2. These results are hard for me to reconcile with those of Grieve et al. In particular, I am concerned that none of the R* values in Figure 3a of Grieve et al. appear to be larger than 0.7. This seems to suggest that Grieve et al. did not consider hillslopes steeper than approximately 0.7*1.2 = 0.84. However, we know that hillslopes steeper than 0.84 are common in OCR. Grieve et al. argue that their results differ from those of Roering et al. due to the different methods for extracting L_H. However, I don't think this adequately addresses the fact that Roering et al.'s slope data clearly show the presence of gradients approaching 1.2 and an increasing planarity of hillslopes as the gradients approach 1.2 in OCR, strongly indicating that Sc is approximately 1.2 in that area. Grieve et al. would likely argue that Sc takes on a range of values, hence the presence of some slopes with gradients above 1.0 or 1.1 does not contradict their conclusion that the average Sc value is 0.79. Maybe this is true, but I would like to see this hypothesis explored in more detail because I find the results of Roering et al. (1999) very convincing in regard to the Sc value they chose.

2) Why might the Grieve et al. approach be flawed enough to provide a misleading measure of Sc and/or an incorrect assessment of steady state? I can think of at least four possibilities. First, the Oregon Coast Range may not be sufficiently in local topographic steady state for their method to apply at the necessary level of precision required for the presence/absence of topographic steady state and the value of Sc to be reliably determined (see, e.g., Sweeney et al., How steady are steady-state landscapes? Using visible–near-infrared soil spectroscopy to quantify erosional variability, Geology, 2012). In particular, their assumption that landscape-scale erosion rates can be extracted from the hilltop curvature seems to assume a topographic steady state and/or a uniformity of erosion rates that may not apply anywhere at the scale they are working. Second, they are applying a 1D model (equation (5)) to 2D reality. This may seem like a quibble, but the fact that there is nothing like a convergent hillslope in their model seems relevant in assessing its ability to definitively allow us to make conclusions regarding relatively subtle aspects of landscape evolution. Third, they assume that colluvial transport flux is independent of soil thickness. If sediment flux is an in-

ESurfD

creasing function of soil thickness (as has been shown by many studies) and soil thickness is lower than the landscape average near divides (also common, since divides are divergent yet hillslopes includes convergent areas where soils tend to be thicker), then the approach of Grieve et al. may systematically overestimate the true value of E* since the value of K will be an underestimate for the landscape as a whole. More broadly, equation (5) is of uncertain applicability if sediment flux is a function of soil thickness, bringing to my mind the question of how confident we can be in the results of this method with regard to the presence/absence of steady state. A fourth possibility is that some fluvial erosion occurs on hillslopes in addition to colluvial erosion. As a result, their model (which includes erosion by colluvial processes only) might underestimate the true erosion rate. I think all of these possibilities should be considered in the analysis or at least acknowledged as possibilities in the revision.

Minor: The paper has a few typos. For example, "couple" should be "coupled" on p. 14, line 19. The publication year of Grieve et al. (2015) should be (2016). All of the references have strange random numbers included after them. These should be removed.

I wish to thank the authors for a stimulating paper and wish them the best as they continue on.

---

## Author Comment (AC1) · 24 Feb 2016

Our response to reviewer 1 is uploaded as three files: our response letter, a copy of the manuscript with our changes highlighted and a clean copy of the manuscript.

Please also note the supplement to this comment:
http://www.earth-surf-dynam-discuss.net/esurf-2015-53/esurf-2015-53-AC1-
supplement.zip

---

## Referee Comment (RC2) · Anonymous Referee #2 · 25 Feb 2016

Grieve and others present a method and software for linking erosion and the relief structure of landscapes. The manuscript is clearly written and the sensitivity analyses provide a useful guide to aid in interpreting the results of their method. Hence, I think this manuscript will be a useful and well-received contribution; the authors have produced a software that will benefit the community. Thank you.

One comment regards the interpretation of data that do not fall on the Roering et al. (2007) curve. In the case of the Oregon Coast Range this could be either because 1) the landscape is in a steady state and the parameterization requires adjustment or 2) the landscape is not in a steady-state and the parameterization is correct. I suspect it is quite difficult to clearly distinguish among these two possibilities, and that both should be discussed in the manuscript. Similarly, for the Coweeta site, there should be some justification for a substitution of the Sc value from the Oregon Coast Range

for the value calculated from the framework, and more explanation of how the field observations indicate alluviation is reducing the critical gradient in this landscape.

An additional comment is that results from this framework shown in Table 1 suggests there is little variation in Sc among the landscapes that were examined, despite differences in the setting of each site. I think this is a discussion point that could be elaborated upon.

The comments and requests from the first reviewer are well-posed and reasonable, and should also be weighed by the authors.

Minor comments: I encourage the authors to search the manuscript for "data is" and replace these instances with "data are".

Page 3. Line 10. "methods published" – this sentence could be rephrased, it is slightly awkward

Page 5. Line 15. Be explicit here about what limits relief. It isn't a critical angle per se, but material strength.

Page 7. Line 12. I suggest inserting a comma following "(2012)"

Page 18. Lines 17-19. It would be useful to show these data in a figure, so that readers can compare results from the two Sc values.

Page 16. Line 21. Replace "This is" with "The high R* values" or similar phrasing to be more explicit.

Figure 5. The ordering and lettering of the panels does not parallel the order in the caption. Check that the main text follows the revised ordering, when referring to this figure.
* * *

---

## Author Response (AR1)

Manuscript prepared for Earth Surf. Dynam.
with version 2015/04/24 7.83 Copernicus papers of the LaTeX class copernicus.cls.
Date: 1 March 2016

**Author's response to reviews of "A nondimensional framework for exploring the relief structure of landscapes" by S. W. D. Grieve et al.**

Stuart W. D. Grieve[1], Simon M. Mudd[1], Martin D. Hurst[2], and David T. Milodowski[1]

[1]School of GeoSciences, University of Edinburgh, Drummond Street, Edinburgh EH8 9XP, UK
[2]British Geological Survey, Keyworth, Nottingham NG12 5GG, UK

*Correspondence to:* Stuart W. D. Grieve (s.grieve@ed.ac.uk)

In this document, reviewers comments are presented in **bold type** and our responses are in standard type. Following our responses to the review comments, a copy of the manuscript with our changes highlighted is included.

**5   Review 1**

**Grieve et al. propose a framework for computing the dimensionless relief and erosion rates of ridge-and-valley topography using a combination of valley network extraction and hilltop curvature analysis. They argue that their approach allows one to determine whether or not a landscape is in topographic steady state and to determine the mean Sc value (the maximum**

10 **gradient of stability) of a landscape. Overall I think the paper will be an excellent contribution, once some issues are thoroughly considered and/or caveats provided.**

We would firstly like to thank the reviewer for their positive and thorough appraisal of our work. We are pleased that the reviewer considers the manuscript and accompanying software to be worth-

15 while and believe that through this discussion the manuscript has been significantly improved. We have added additional discussion into the manuscript to account for the limitations highlighted by the reviewer, have enhanced the theoretical background to place the work in a better context and have added additional detail to the methodology to clarify how our data is extracted. We respond to each point made by the reviewer individually below.

**(1) I am skeptical that the mean value of Sc is $0.79$ in the Oregon Coast Range. Roering et al. (1999) demonstrated that many hillslopes in the OCR have gradients in the 0.8 to 1.1 range, and more importantly that the planarity of hillslopes systematically increases as gradients approach 1.2. These results are hard for me to reconcile with those of Grieve et al. In**

particular, I am concerned that none of the $R^*$ values in Figure 3a of Grieve et al. appear to be larger than 0.7. This seems to suggest that Grieve et al. did not consider hillslopes steeper than approximately $0.7 * 1.2 = 0.84$. However, we know that hillslopes steeper than 0.84 are common in OCR. Grieve et al. argue that their results differ from those of Roering et al. due to the different methods for extracting $L_H$. However, I don't think this adequately addresses the fact that Roering et al.'s slope data clearly show the presence of gradients approaching 1.2 and an increasing planarity of hillslopes as the gradients approach 1.2 in OCR, strongly indicating that Sc is approximately 1.2 in that area. Grieve et al. would likely argue that Sc takes on a range of values, hence the presence of some slopes with gradients above 1.0 or 1.1 does not contradict their conclusion that the average Sc value is 0.79. Maybe this is true, but I would like to see this hypothesis explored in more detail because I find the results of Roering et al. (1999) very convincing in regard to the Sc value they chose.

As detailed in Grieve et al. (2016), the critical gradient which we constrain using these techniques is necessarily an averaged value, as the fitting procedure will pass the steady state curve through the center of the cluster of data points. This results in a critical gradient where approximately 50% of the hillslopes will exceed the best fit $S_c$. Other attempts to constrain $S_c$ have been made using similar best fit methods (DiBiase et al., 2010; Hurst et al., 2012, 2013) which have found similar underestimates in landscapes when contrasted with the value used by Roering et al. (1999). As noted by the reviewer we argue that our best fit $S_c$ value is the average value of a probability distribution of critical gradients, whereas the value of Roering et al. (1999) can better be considered as an upper bound $S_c$ and should that value be required in a study, the methods of Roering et al. (1999) would be better employed. Our method has the advantage of requiring little user supervision and no field data to constrain an average $S_c$ value and as such can facilitate rapid, broad scale comparisons between landscapes.

We have rewritten the discussion of these points in the manuscript in order to better convey our distinction between different definitions of $S_c$, in particular we have explicitly made the distinction between the techniques for extracting $S_c$ and have outlined why we observe differences between the results of the methods. The explicit comparison between the $S_c$ values of the 2 methods for the Oregon Coast Range and Gabilan Mesa has also been removed from the discussion, to allow the intercomparison of different best fit methods, rather than the comparison between an average and a maximum value being made. We have also added a sentence at the end of the discussion to urge the reader to consider which value of $S_c$ is suitable for a given study.

In particular, I am concerned that none of the $R^*$ values in Figure 3a of Grieve et al. appear to be larger than 0.7. This seems to suggest that Grieve et al. did not consider hillslopes steeper than approximately $0.7 * 1.2 = 0.84$. However, we know that hillslopes steeper than 0.84 are

**common in OCR. Grieve et al. argue that their results differ from those of Roering et al. due to the different methods for extracting $L_H$.**

65   In the original Roering et al. (2007) work, the upper limit of $R^*$ calculated for the Oregon Coast Range was approximately $0.8$, which yields a maximum gradient of $0.96$, or a difference between these two gradients of $0.12$. Aside from the previously highlighted difference in methods for measuring $LH$ and $R$ between the two works, which result in a small increase in $R$ and a larger increase in $LH$ (which has a net result of decreasing $R^*$), two other factors will influence the observed $R^*$ val-

70   ues. Firstly, the spatial averaging will act to dampen the extreme values (both high and low), which can be seen by looking at the changes in the range of data points between Figure 5B and 5C and secondly, as demonstrated by Hurst 2012, any hilltops with a gradient above $0.4$ must be excluded in order to use hilltop curvature as a proxy for erosion rate.

75   **(2) Why might the Grieve et al. approach be flawed enough to provide a misleading measure of Sc and/or an incorrect assessment of steady state? I can think of at least four possibilities. First, the Oregon Coast Range may not be sufficiently in local topographic steady state for their method to apply at the necessary level of precision required for the presence/absence of topographic steady state and the value of Sc to be reliably determined (see, e.g., Sweeney et**

80   **al., How steady are steady-state landscapes? Using visible–near-infrared soil spectroscopy to quantify erosional variability, Geology, 2012). In particular, their assumption that landscape-scale erosion rates can be extracted from the hilltop curvature seems to assume a topographic steady state and/or a uniformity of erosion rates that may not apply anywhere at the scale they are working.**

85

   In this paper we employ the same definition of steady state as was used in Grieve et al. (2016), whereby we consider a hillslope which retains a constant topographic form in relation to its baselevel, the channel at the base of the hillslope, to be in steady state. Such a formulation, defined by Mudd and Furbish (2004), and employed on a range of transient landscapes by Hurst et al. (2012, 2013)

90   within the context of $E^* R^*$ analysis, allows for the extraction of erosion rates across diverse tectonic and erosional regimes using hilltop curvature.

   The clustering of the values in $E^* R^*$ space when contrasted to transient landscapes such as Cascade Ridge (Figure 4) give us confidence that the Oregon Coast Range is in approximate steady state. We consider the variability which we observe in the $E^* R^*$ data in such a landscape to high-

95   light the catchment scale variability which has been predicted in models (Reinhardt and Ellis, 2015) and elegantly demonstrated with field data by Sweeney et al. (2012).

   We have updated the theory section to include this explicit definition of steady state with the aim of increasing the clarity of our later discussions surrounding landscapes which are in steady state,

such as the Oregon Coast Range.

100

**Second, they are applying a 1D model (equation (5)) to 2D reality. This may seem like a quibble, but the fact that there is nothing like a convergent hillslope in their model seems relevant in assessing its ability to definitively allow us to make conclusions regarding relatively subtle aspects of landscape evolution.**

105

The challenge of applying 1D models to 2D reality is one which we should have explicitly addressed in this manuscript. The data generation and topographic analysis is designed to ensure that we are best able to apply these 1D models to our 2D data by following the methodology of Hurst et al. (2012), who first encountered this challenge. The valley extraction algorithm we employ, which is designed after Pelletier (2013) and Passalacqua et al. (2010) identifies areas of high convergence on hillslopes as valleys, which, when used as the end points for the flow routing algorithm employed to generate $L_H$ and $R$ measurements effectively exclude hillslope traces which cross convergent topography. We have added to the theory section of the paper to address this factor explicitly, presenting a brief review of methods of applying 1D models to 2D topography, and directing the reader to consider our results within the context of the challenges inherent in analyzing real topographic data.

**Third, they assume that colluvial transport flux is independent of soil thickness. If sediment flux is an increasing function of soil thickness (as has been shown by many studies) and soil thickness is lower than the landscape average near divides (also common, since divides are divergent yet hillslopes includes convergent areas where soils tend to be thicker), then the approach of Grieve et al. may systematically overestimate the true value of E\* since the value of K will be an underestimate for the landscape as a whole. More broadly, equation (5) is of uncertain applicability if sediment flux is a function of soil thickness, bringing to my mind the question of how confident we can be in the results of this method with regard to the presence/absence of steady state.**

The reviewer correctly highlights that there is evidence for depth dependent sediment transport on many hillslopes (e.g., Braun et al., 2001). Roering (2008) performed a non-dimensionalization of sediment transport from depth dependent creep and demonstrated an increased sensitivity of hilltop curvature to erosion rates under this transport regime. Which could be used to account for an increase in $E^*$ values in a landscape. Unfortunately it is not currently possible to perform large scale experiments using real topography incorporating a depth-slope product flux law as we do not have soil thickness information at the spatial scale required.

135    Grieve et al. (2016) used topography to falsify predictions of the linear and nonlinear sediment flux models, but was unable to make any predictions regarding other sediment flux models due to either a lack of analytical solutions to the models, or as is the case for the depth-slope product, a lack of soil thickness information. The resulting work demonstrated that topography in four fieldsites was consistent with a nonlinear sediment flux law and this is the basis with which we employ such a
140    model in this study, which shares the same fieldsites.

In light of this clear limitation to our results we have extended the Theoretical Background section to discuss the existence of other flux laws, and the basis of our selection of the nonlinear model over any of the other choices. We do not consider it possible to quantify the uncertainty in our results as we have no constraint on soil thickness, but trust that by drawing readers to this limitation near the
145    start of the manuscript we can avoid any misunderstanding of the applicability of both our results and the technique as a whole.

**A fourth possibility is that some fluvial erosion occurs on hillslopes in addition to colluvial erosion. As a result, their model (which includes erosion by colluvial processes only) might un-**
150    **derestimate the true erosion rate. I think all of these possibilities should be considered in the analysis or at least acknowledged as possibilities in the revision.**

This is a fundamental observation which impacts upon many facets of topographic analysis. In the case of this study we have taken great care to employ a valley extraction scheme based on the work
155    of Pelletier (2013) and Passalacqua et al. (2010) which performs well across a range of landscapes in defining the transition point between channel and hillslope. However, no such algorithm is perfect when applied to real topographic data and as such we have expanded our methods section to more explicitly address this particular limitation of our topographic analysis.

160    **Minor: The paper has a few typos. For example, "couple" should be "coupled" on p. 14, line 19. The publication year of Grieve et al. (2015) should be (2016). All of the references have strange random numbers included after them. These should be removed.**

We have corrected these typos and a small number of other mistakes which were noticed during
165    the typesetting process. The strange random numbers included after the references appear to have been added during the typesetting process, but we are investigating how to ensure that they do not appear in the final manuscript.

**I wish to thank the authors for a stimulating paper and wish them the best as they continue**
170    **on.**

**Review 2**

**Grieve and others present a method and software for linking erosion and the relief structure of landscapes. The manuscript is clearly written and the sensitivity analyses provide a useful guide to aid in interpreting the results of their method. Hence, I think this manuscript will be a useful and well-received contribution; the authors have produced a software that will benefit the community. Thank you.**

We would like to thank the reviewer for their detailed consideration of our manuscript. We are pleased that the reviewer sees merit in the software we have produced, and agree that our sensitivity analyses should help the community to better interpret $E^*R^*$ measurements. Following the recommendations of this review we have modified Figure 5 to correct an error in how we labeled the subplots and have added a new figure to aid with the interpretation of the Coweeta results. We have expanded our discussion in several places to allow for possible alternative explanations of our results and have also implemented all of the minor edits suggested. We believe that these changes have significantly improved the quality and clarity of our manuscript. We respond to each point made by the reviewer individually below.

**One comment regards the interpretation of data that do not fall on the Roering et al. (2007) curve. In the case of the Oregon Coast Range this could be either because 1) the landscape is in a steady state and the parameterization requires adjustment or 2) the landscape is not in a steady-state and the parameterization is correct. I suspect it is quite difficult to clearly distinguish among these two possibilities, and that both should be discussed in the manuscript.**

The reviewer is correct that it is challenging to distinguish between these possibilities purely from topographic data. However, there is a considerable amount of evidence for the Oregon Coast Range being in steady state (e.g., Reneau and Dietrich, 1991; Roering et al., 2007) and the selection of a lower average $S_c$ value is supported by earlier work to constrain the critical gradient (Grieve et al., 2016). We have added a sentence to the discussion of the Oregon Coast Range results to highlight this possible alternative interpretation: *'This can be interpreted as evidence for topographic decay, however due to the preponderance of evidence supporting a steady state hypothesis for this landscape (e.g., Reneau and Dietrich, 1991; Roering et al., 2007), it is also possible that a critical gradient of 1.2 is too large in this location.'*

In responding to reviewer 1 we have also addressed these points more broadly, by expanding on our theory section and explicitly stating our definition of steady state. The challenges of parameterizing the model and fitting the critical gradient are now discussed in more detail in the manuscript,

with the aim of drawing the reader's attention to the potential limitations inherent within this method.

210 **Similarly, for the Coweeta site, there should be some justification for a substitution of the Sc value from the Oregon Coast Range for the value calculated from the framework,**

Our selection of this value is based on the similarities in landscape morphology, sediment transport processes and the range of $E^*$ values observed at the two sites. The aim of this selection of a
215 critical gradient is not to suggest that this is the correct value for Coweeta, but rather that a larger critical slope than the value 0.57 of reported in Grieve et al. (2016) produces data consistent with topographic decay. We have rewritten this section to better reflect that we do not claim that 0.79 is the correct value, rather that it is within a more plausible range than the value of 0.57.

220 **and more explanation of how the field observations indicate alluviation is reducing the critical gradient in this landscape.**

When we referred to the mean gradient reducing we should have instead referred to the mean relief reducing, which has the consequence of reducing the gradient. In locations where valley alluviation
225 is occurring, the base of the hillslope is raised vertically with regard to the elevation of the ridgeline. This reduces the relief whilst not changing the morphology of the hillslope, due to the difference in timescales of hillslope and channel response (Hurst et al., 2012). This can result in a reduction in $R^*$ values across a landscape, leading to a lower $S_c$ value being generated for a landscape than would be predicted were the alluviation not taking place. We have clarified this position and added
230 a more detailed explanation of this mechanism: *'As a valley fills with sediment, the hillslope relief will be reduced more rapidly than other hillslope properties, due to the difference between rates of hillslope and channel response to forcing (Hurst et al., 2012). Such a reduction in relief will reduce $R^*$, resulting in a reduced best fit $S_c$ value.'*

235 **An additional comment is that results from this framework shown in Table 1 suggests there is little variation in Sc among the landscapes that were examined, despite differences in the setting of each site. I think this is a discussion point that could be elaborated upon.**

We have added to the discussion to raise this point, highlighting that it is a function of the ranges
240 of E* and R* values we observe in these landscapes, and that similar studies have arrived at the same magnitude of values for critical gradient. The following paragraph has been added to the critical gradient section: *'The similarity of the average $S_c$ values obtained using the bootstrapping procedure across three diverse landscapes highlights the presence of a distribution of $E^* R^*$ values existing for each landscape, and the nature of an average SC measurement. Such a distribution occurs due*

245 *to local variations in topography, process and material properties and similarities can be drawn between the results presented in Table 1 and other similar studies (DiBiase et al., 2010; Hurst et al., 2012).'*

**The comments and requests from the first reviewer are well-posed and reasonable, and**
250 **should also be weighed by the authors.**

We have made significant alterations to the manuscript following the recommendations of reviewer 1, details of which can be found in our response to that review.

255 **Minor comments: I encourage the authors to search the manuscript for "data is" and replace these instances with "data are".**

We have made the requested changes.

260 **Page 3. Line 10. "methods published" – this sentence could be rephrased, it is slightly awkward**

We have rephrased this sentence to better convey our meaning: *'Such fundamental relationships provide important insight into landscape evolution, however many of these techniques are challeng-*
265 *ing to implement, due to variable or poorly defined methods, or require proprietary software to obtain data.'*

**Page 5. Line 15. Be explicit here about what limits relief. It isn't a critical angle per se, but material strength.**
270
We have rephrased this section to highlight that the critical angle is controlled by material strength.

**Page 7. Line 12. I suggest inserting a comma following "(2012)"**

275 Done.

**Page 18. Lines 17-19. It would be useful to show these data in a figure, so that readers can compare results from the two Sc values.**

280    We have produced an additional figure (Figure 6 in the new manuscript) which displays this data and have incorporated the figure into our discussion of potential topographic decay in this location.

**Page 16. Line 21. Replace "This is" with "The high R\* values" or similar phrasing to be more explicit.**

285

Done.

**Figure 5. The ordering and lettering of the panels does not parallel the order in the caption. Check that the main text follows the revised ordering, when referring to this figure.**

290

The main text in the manuscript is consistent with the figure caption, and this was the intended ordering of the subplots. Consequently we have re-generated Figure 5 with the subplots in the correct order to reflect the text and the caption.

[revised manuscript text omitted]

---

## Editor Decision (ED1)

[revised manuscript text omitted]
                       | 6.55                              | $0.07\pm0.03$                                         | 0.06                    |
| Gabilan Mesa                             | 5.56                              | $0.20\pm0.15$                                         | 0.11                    |
| Cascade Ridge                            | 9.84                              | $0.17\pm0.13$                                         | 0.11                    |
| Coweeta                                  | 8.91                              | $0.17\pm0.13$                                         | 0.11                    |
| Gabilan Mesa
Cascade Ridge
Coweeta | 5.56
9.84
8.91              | $0.20 \pm 0.15$
$0.17 \pm 0.13$
$0.17 \pm 0.13$ | 0.11
0.11
0.11    |

Table A1. LiDAR point cloud metadata.

[revised manuscript text omitted]

---

## Author Response (AR2)

Manuscript prepared for Earth Surf. Dynam.
with version 2015/04/24 7.83 Copernicus papers of the LaTeX class copernicus.cls.
Date: 17 March 2016

**Response to associate editor comment on "A nondimensional framework for exploring the relief structure of landscapes" by S. W. D. Grieve et al.**

Stuart W. D. Grieve[1], Simon M. Mudd[1], Martin D. Hurst[2], and David T. Milodowski[1]

[1]School of GeoSciences, University of Edinburgh, Drummond Street, Edinburgh EH8 9XP, UK
[2]British Geological Survey, Keyworth, Nottingham NG12 5GG, UK
*Correspondence to:* Stuart W. D. Grieve (s.grieve@ed.ac.uk)

In this document, the associated editor's comments are presented in **bold type** and our responses are in standard type. Following our responses to the review comments, a copy of the manuscript with our changes highlighted is included.

5 **The authors have adequately addressed the reviewer's comments and the paper should be recommended for publication pending some minor revisions, outlined below.**

We would like to thank the associate editor for these additional comments on our manuscript, in making these changes we believe that the paper will have a broader impact. We have corrected the 10 errors in the text and references to figure subplots and have added a new Figure 2 to give an overall view of the relationship between the differing spatial units described in the manuscript.

**Firstly and most importantly, because this special issue encompasses a readership from many domains, I think it would be useful for the authors to include one or two more figures 15 which illustrate the concepts and procedures in this contribution. For example, for one of their study areas they could include a map-view illustration of the delineation of basins, hillslope patches, channel networks, hilltop points and their associated flowpaths, etc. Another example could be a map-view of the hillslopes/basins included/excluded in by the filtering in 6.2.2. These are suggestions and I leave the authors to choose the most appropriate figure(s) to illustrate 20 their methodology.**

A new figure (Figure 2 in the new manuscript) has been produced which shows example hillslope flow paths, hilltop patches, and basins for a section of Gabilan Mesa, with the aim of providing the reader with a clearer understanding of the areal units used in the spatial averaging processes 25 discussed in the manuscript. The display of flow paths in this figure also gives the reader a clearer

understanding of how the key data is generated for this paper. The new figure also displays hilltop patches of varying size, highlighting the outcomes of our hilltop identification algorithm.

**In implementing reviewer 2 change to d̈ata are, the authors have neglected to adopt the plural in their other uses of the word d̈ata, e.g. ẗhe data plots, rather than ẗhe data plot, I have tried to outline this in the commented-up PDF, but encourage the authors to double-check for consistency themselves.**

We have made the requested changes and have checked for and corrected any further inconsistencies arising from our implementation of the second reviewer's comments.

**I have spotted that in two of the figures, the plots appear in an order which is different to that outlined in the caption and the text (Figs.8 9), as this issue was outlined also by reviewer2 for another figure, I encourage the authors to make sure this ordering-issue is fixed in the final version.**

We have amended the text to correctly refer to Figure 8 (now Figure 9), and have reordered the subplots of Figure 9 (now Figure 10) so that they are correctly referred to throughout the text.

**Other minor suggestions are given in the commented PDF.**

We have followed the guidance provided within the commented PDF to fix some typos and regenerate Figure 1 to make better use of the space by compressing trailing digits in the UTM coordinates.

[revised manuscript text omitted]